# A Generalization Performance Study Using Deep Learning Networks in Embedded Systems

**DOI:** 10.3390/s21041031

**Published:** 2021-02-03

**Authors:** Joseba Gorospe, Rubén Mulero, Olatz Arbelaitz, Javier Muguerza, Miguel Ángel Antón

**Affiliations:** 1TECNALIA, Basque Research and Technology Alliance (BRTA), Astondo Bidea Building 700, 48160 Derio, Spain; ruben.mulero@tecnalia.com (R.M.); mangel.anton@tecnalia.com (M.Á.A.); 2Electronics and Computer Science Department, Mondragon Unibertsitatea, Loramendi 4, 20500 Arrasate-Mondragon, Spain; 3Faculty of Informatics, University of the Basque Country UPV/EHU, Manuel Lardizabal 1, 20018 Donostia, Spain; olatz.arbelaitz@ehu.eus (O.A.); j.muguerza@ehu.eus (J.M.)

**Keywords:** edge computing, deep learning, quantisation, computer vision

## Abstract

Deep learning techniques are being increasingly used in the scientific community as a consequence of the high computational capacity of current systems and the increase in the amount of data available as a result of the digitalisation of society in general and the industrial world in particular. In addition, the immersion of the field of edge computing, which focuses on integrating artificial intelligence as close as possible to the client, makes it possible to implement systems that act in real time without the need to transfer all of the data to centralised servers. The combination of these two concepts can lead to systems with the capacity to make correct decisions and act based on them immediately and in situ. Despite this, the low capacity of embedded systems greatly hinders this integration, so the possibility of being able to integrate them into a wide range of micro-controllers can be a great advantage. This paper contributes with the generation of an environment based on Mbed OS and TensorFlow Lite to be embedded in any general purpose embedded system, allowing the introduction of deep learning architectures. The experiments herein prove that the proposed system is competitive if compared to other commercial systems.

## 1. Introduction

In recent years, manufacturing industries have become increasingly involved in the improvement of their value chain by digitalising their processes. This implies the inclusion of new key elements that create new ways of understanding how processes work and how they can be optimised by making real-time changes. Concepts such as the Internet of Things (IoT), Big Data or artificial intelligence (AI) are paving new ways in the creation of a closer connection between the automation of manufacturing processes and experts. Thus, it is possible to obtain real-time information from different sensors deployed in the plant that have the ability to perform critical actions when required.

However, these actions need to be taken by a decision-making process that requires the knowledge from an expert in the field. These experts should define the required conditions to perform accurate actions. The main problem is that these processes need to be automated to ensure that the actions are made at the right moment; thus, the decision-making process should be automated by an intelligent agent with the capacity of knowing when the action should be executed. This can only be achieved by using two different approaches: (1) a rule-based approach (expert systems) [1], where an expert defines which actions are required based on different conditional values; (2) a machine learning (ML) [2] approach, where a mathematical algorithm is trained using data captured from a set of sensors to predict the possible outcomes and to execute the needed actions. The former is feasible when the problem is simple and does not require several actions, while the latter is convenient when the problem is more complex and requires several variables to evaluate the best action to take. Thus, the inclusion of AI-based algorithms is becoming the backbone of several industrial processes able to make automatic decisions in the same way an expert could, based on previously recorded data.

ML is a subset of AI that comprises several mathematical algorithms to create intelligent agents with the capability of making predictions or classifications. The training process evaluates the outputs given by the algorithms with the real outputs given by a labelled data set (training set) used as a proof of its capacity for making the right predictions. To make this possible, it is necessary to use data samples labelled by an expert and a powerful machine such as a personal computer (PC) or a high-performance computer (HPC) to fit the mathematical algorithms in an interactive manner. The most advanced ML-based algorithms, called deep neural networks (DNNs) or deep learning (DL) [3], require powerful graphic cards (graphic processing units (GPUs)) and several training hours to produce an accurate decision-making agent. Different approaches [4,5] rely on using a centralised solution where an HPC is capable of managing different data sources, of executing ML algorithms, and of making the required actions to correct or optimise the execution plan that is used. However, there are processes where this type of architecture is not feasible because critical actions need to be taken as fast as possible without any delay in the execution pipeline.

In this regard, a centralised architecture based on an HPC is not capable of guaranteeing that actions can be executed on time. The reason is that the communication among the HPC, sensors, and actuators requires a set of different connections, protocols, and bandwidths to gather, evaluate, and execute the actions. Thus, the only solution to overcome this problem and to have a direct response is by improving the response time and saving the connection bandwidth. This can only be achieved by bringing data gathering, decision-making, and actuator processes to the edge; hence, the concept of edge computing (EC) [6] has emerged in the literature.

EC comprises the required techniques to have devices with the capacity of taking actions based on the data captured by the device itself, making it reliable to take critical actions without requiring the usage of an HPC solution. These actions do not need to wait until a resolution is achieved, reducing the usage of external services and the connection bandwidth. The actions can be intelligent if an ML-based model is added to the device, i.e., if the model is trained in an external HPC machine and then added into the EC device. However, EC devices have small computing capabilities and small memory capacities; therefore, they are not able to achieve the same performance that an HPC solution could. The most advanced algorithms are computationally expensive and require large sets of space and memory to provide accurate results.

The results obtained from an EC device are even worse when the applied ML algorithm is too complex, e.g., DNNs, because there are not enough computing resources, memory, or disk space to execute it properly. In fact, DNNs are not prepared to be fully executed in EC devices because they are designed to be executed in PCs or HPCs without taking into account any computing limitations. Thus, the process of fitting a DNN model into an EC device is a twofold open challenge. On the one hand, it is necessary to fit the DNN model inside a low-resource device, but in addition, the results obtained with the compressed model must be as accurate as the ones obtained with the complete model.

In this context, several authors have tried to provide different approaches to reduce the size of ML algorithms with the aim of fitting them into the device without losing their prediction accuracy. However, at this moment, the predictions obtained by these reduced models are far from being as good as the ones deployed in an HPC solution. In addition, the process of fitting the models into an EC device is not straightforward, because embedded system manufacturing companies do not have a generalised method to do so. Each one provides its set of tools and operating systems to fit an algorithm into the embedded system (for example, ST Microelectronics has its X-Cube-AI expansion pack (https://www.st.com/en/embedded-software/x-cube-ai.html (accessed on 1 February 2021))). In this regard, there are free open-source implementations such as ARM Mbed OS (https://os.mbed.com/mbed-os/ (accessed on 1 February 2021)), which tries to collect different embedded systems from different manufacturers to run programs inside it, but, at this moment, in terms of ML algorithms, they lack a standardised way of doing this.

This implies that even if authors are capable of developing new ways of compressing algorithms to fit them into EC devices, there is still an open challenge, because: (1) there is not a standardised methodology to implement and execute them inside an EC device; (2) the real performance of a compressed DL algorithm is unknown, and the only way of knowing it is by executing it once it has been implemented on the target device; (3) there is no real information of the performance levels of the different EC devices; (4) the open/closed solutions do not provide in depth information about the techniques to use to compress a DL network and its final performance. For that reason, having a standardised way of executing ML algorithms in an embedded system provides the opportunity to check: (1) the generalisation capacity of executing ML algorithms over different EC devices; (2) understanding how the implemented algorithms perform in different machines; (3) which improvements from the state-of-the-art could be used in order to obtain fitted ML algorithms with the capacity of giving accurate results.

In this paper, we present novel research in which we compare the performance of a DL algorithm in different devices with different compression (fitting) techniques to study which elements affect the performance of the applied ML algorithm. To do so, we propose the usage of a highly-demanding DNN model, whose aim is, provided an image, to determine whether a person appears in it or not and to study the performance loss when the algorithm is altered by the most recent compression techniques. In addition, we propose a development environment based on Mbed OS and TensorFlow Lite in order to achieve a homogeneous DL network integration procedure in embedded systems. Furthermore, we compare these results with different commercial boards that claim to obtain high accuracy results in their commercial closed-source environments. To perform these experiments, we use the open operating system called ARM Mbed OS and a computer vision algorithm based on DNNs called Google MobileNet [7] to understand how the current compression techniques affect the results.

Thus, our contributions in this paper are:A comparison of the different techniques to reduce the size of DL algorithms.A performance comparison of the different methods on a single case of human detection using a standalone PC and an EC device.A proposal and performance analysis of a generalist development environment for integrating DL networks into embedded systems.

The remainder of this paper is divided as follows: Section 2 contains the state-of-the-art about the current techniques to reduce the size of an ML-based algorithm. Section 3 contains the experimental procedure carried out in this paper to demonstrate the performance of the different methods. Section 4 states all of the experiments carried out, and Section 5 details the results obtained and their discussion. Finally, Section 6 contains the conclusions of this study.

## 2. State-of-the-Art

### 2.1. Deep Learning

DL consists of both supervised and unsupervised machine learning algorithms based on artificial neural network (ANN) [8] layers that try to represent complex architectures, e.g., the human brain, a complex industrial machine, or the structure of sentences and idioms. ANNs are composed of neurons that form layers with the capacity to process the complex inputs of data to produce a simplified output.

Figure 1 (extracted from (trends.google.es/trends/ (accessed on 1 February 2021))) depicts the evolution of the growth of DL usage with respect to other decision-making system techniques. In spite of DL being the most used ML algorithm, its first usage was proposed back in the 1990s [9]. Due to the low computational capacity of the computer systems at that time, it was not feasible to train these algorithms. Currently, however, the problem is addressed with the advantage of the current technology based in the usage of GPU devices, which makes capable the training of complex ANNs in a reasonable time. Therefore, the trend of DL has recently emerged [10].

Not only have the computational capabilities improved, but also the ANNs themselves and the way they are trained. In this regard, there are new activation functions such as rectified linear units (ReLUs) [11], sigmoid linear units (SiLUs) [12], or exponential linear units (ELUs) [13]. In addition, there are different methods to initialise network weights, such as the ones presented in [14]. Furthermore, different types of neurons have been proposed for DL networks, each with different purposes. For example, we can highlight convolutional neurons, recurrent neurons, or long short-term memory (LSTM) neurons. With these neurons, new network structures have been proposed with the aim of improving specific aspects. For example, convolutional neural networks (CNNs) are able to extract relevant features from images and videos automatically, avoiding the need of specifically extracting the features from them. Recurrent neural networks (RNNs) allow past actions to be considered while making decisions in the present, which can be very useful for analysing the context of a word in a complete sentence.

The diversity of DL architectures makes possible the implementation of a wide range of applications in different research fields. Applications include predicting new composites for antibacterial compounds [15] in medicine, detecting intrusion attacks [16] in cybersecurity, exploring written messages by creating applications based on natural language processing (NLP) [17], or obtaining fine-grained information from a single image using computer vision (CV) [18]. CV is one of the fields where deep learning has been used so far, as neural networks can extract features from images, making the image processing procedure much easier. One of its applications is object recognition for applications such as traffic sign detection for advanced driver assistance systems [19], fruit recognition systems to optimise harvesting in the agricultural environment [20], quality analysis of materials for manufacturing processes [21], 3D modelling systems to obtain the structure of the objects based on different images [22], etc.

All of these applications indicate that there is work to be done in this research field. The usage of DL with EC has a wide spectrum of contexts and applications, which creates a potential interest in the industrial area to provide accurate solutions to the end users.

### 2.2. Edge Computing

As mentioned in the previous section, the scope of EC comes from the need to run applications without the requirement of a complex architecture to transmit the information to an external server. In order to find potential solutions to avoid the externalisation of information, different alternatives have been proposed in the scope of EC. Besides integrating the intelligence into the final node, intermediate approaches have been considered where the idea of making the decision as close as possible to the action is combined with outsourcing certain services to a server. Some of these architectures are summarised below:
Cloudlet and micro data centres (MDCs): This type of architecture is composed of three levels. First, they maintain the central server or what can also be called the cloud. Second, they have an intermediate network, or a micro cloud, which is located closer to the decision-making process and has less capacity than the cloud. Third, the final devices are used to offer different services, such as video on demand or connectivity between mobile devices. This type of service usually requires much storage space and fast connections. By distributing the load between different levels, it is possible to create a main storage in the cloud and a secondary storage closer to the client, so that the service can be provided in a reasonable time.Fog computing: This term refers to a totally distributed architecture, where the levels are not defined directly, but rather, a network is proposed where all of the devices are connected to each other, both the end nodes and powerful devices like servers. The idea of having all devices connected among them is also called the IoT. In this case, fog computing is focused on implementing applications where decisions have to be made in real time, rather than offering services closer to the client.Mobile (multi-access) edge computing (MEC): MEC is a standard architecture defined by the European Telecommunications Standards Institute (ETSI), which is installed in the base centres of the telephone network, allowing providing services similar to Cloudlet and MDC. This avoids the requirement of creating an architecture because it uses the stations that are already installed.

Depending on the design of these architectures, it is possible to differentiate the presented systems into three types: (1) cloud intelligence, (2) edge intelligence, and (3) on-device intelligence. Cloud intelligence is the system with the highest computational capacity, allowing making complex decisions in several scenarios, but these decisions increase the latent capacity. On-device intelligence allows decisions to be made directly on the device that is capturing the data, which provides the possibility of acting immediately with a low latency. In these cases, the capacity of the devices is usually very limited, making it difficult to integrate the decision-making system into it. Finally, edge intelligence is an intermediate point between both, where the decision-taking system is directly connected to the final node, but not integrated into it.

This paper focuses on on-device intelligence, where the combination with DL is considered to allow the integration of robust decision-making systems.

### 2.3. Optimisation of DL Networks

Compression techniques are a crucial part of integrating DL networks into embedded systems due to the CPU and memory restrictions. Model optimisation not only allows the reduction of the network size, but also the reduction of its latency. Current model optimisation techniques are divided into four main categories: (1) parameter pruning and quantisation, (2) low-rank factorisation, (3) transfer/compact convolutional filters, and (4) knowledge distillation.

#### 2.3.1. Parameter Pruning and Quantisation

These types of techniques were originally intended to avoid the adjustment of the models to the training data set (overfitting), achieved by reducing the structure of the network in order to obtain a better generalisation capacity. Later, the study was further developed to include network compression to improve the reduction of the network structure. Thus, this technique can be divided in two different parts:
Pruning: This focuses on eliminating those neurons that are lighter or less representative, improving the generalisation of the model and decreasing its size and computational costs. This technique was proposed back in 1990 [23], where the authors pruned a large ANN by estimating the sensitivity of the error function for each connection and eliminating the lowest ones. There are several methods for pruning a network, e.g., it is possible to prune a network based on the structure, instead of pruning individual parameters [24]. Other methods such as those in [25,26] consider grouping parameters based on the hardware or software of the system to be integrated. Another common method is to prune parameters based on the score of different metrics (e.g., the sensitivity of the global error [27]). Different techniques also consider different scheduling approaches, where some methods prune all of the given parameters in a single step [28], while other methods prune them iteratively [29], and some even vary the pruning rate of each iteration depending on complex functions [30]. Some techniques use fine-tuning methods, the most common of which is to continuously train the network by performing iterations. Finally, others perform a network reset [31] or simply a network rewind to an earlier state [28].Quantisation: This attempts to reduce the number of bits needed to represent a parameter, maintaining the accuracy of the network. It can be used both to avoid overfitting and to reduce the size of the network. The first proposal of a practical implementation of the quantisation technique was made in 2014 [32], implementing three different algorithms: (1) k-means clustering, (2) product quantisation [33], and (3) residual quantisation [34]. Quantisation into one or two bits was also analysed in [35], where the authors proposed a quantisation technique to apply during the training of the network, which allows reducing the accuracy loss of the quantised network.

To date, promising results have been achieved by combining different types of techniques. For example, the compression learning by in-parallel pruning-quantization (CLIP-Q) [36] method achieves compression ratios up to 72-times lower than the original network, and the accuracy can be improved by up to 0.7%. To do this, it combines quantisation and pruning techniques, applying them iteratively at each step of the training.

#### 2.3.2. Low-Rank Factorisation

Low-rank factorisation techniques focus on reducing the computational cost of CNNs. The convolution operator multiplies the input tensors of the network with a matrix called “filter”. These filters try to extract the underlying structure of the given tensor to detect objects inside an image or logical connections in time series data. However, the convolutional operations are computationally demanding, which makes them unfeasible for use in tiny environments such as EC devices. For this reason, low-rank factorisation focuses on decomposing the convolutional filters of the network into smaller filters to avoid redundant operations. Figure 2 depicts a filter (represented by an n×m matrix) that can be decomposed into two smaller filters (i.e., n×r and r×m). These filters are able to increase the inference speed of the model and to decrease the number of parameters of certain filters. Some filters are able to decrease other filters with a ratio of up to 13-times less [37], where others are able to compress the size of the base network by almost 50% [38].

For these techniques to be effective, it is necessary to train the model intensively, since by breaking down the filters, the model takes longer to converge to the best solution.

#### 2.3.3. Transferred/Compact Convolutional Filters

These techniques are focused on designing DL networks by trying to reduce the size and computational cost of convolutional filters. To this end, they apply specific transformations to the filters, taking into account how the network behaves. This technique is based on the theory of [39] (Equation (Equation 1)), which says that it is the same to first apply a transformation $T^{\prime }$ to the input (*x*) of a filter Φ and then filter it, as it is to first filter the input and then transform it. In this case, the $T^{\prime }$ and T transformations do not necessarily need to be the same, as they operate on different objects (Φ(x) and *x*).
(1)$$T^{\prime }\Phi \left(x\right)=\Phi \left(Tx\right)$$

Currently, several DL networks have been designed taking into account transfer/compact convolutional filters. The most noticeable ones are MobileNet, which has three different versions [7,40,41]), or SqueezeNet [42], where the common 3×3 convolutional filters are replaced by 1×1 convolutional filters, making the networks more compact. These networks are capable of obtaining highly accurate results while maintaining a low network size.

The main problem with these solutions is that they must be considered in the design of the network; they cannot be applied during or after training, as the architecture of the network must be modified. This implies that it is necessary to have deep knowledge of the original structure of the DL architecture.

#### 2.3.4. Knowledge Distillation

Knowledge distillation is related to knowledge transfer, where the knowledge obtained from one network is transferred to another. In the case of knowledge distillation [43], a base model is set as the teacher and a more compact version of that network as the student. Being so, the training of the compact network is not focused on solving the problem of having the original inputs and outputs, but rather on copying the operation of the base network. It has also been shown that this concept can be transferred to neurons, where a group of significant neurons can play the role of the teacher [44]. These techniques are not compatible with all types of ANNs and often have difficulties in achieving results as good as those obtained using the other methodologies explained above.

Among these optimisation techniques, quantisation is analysed and tested herein. As mentioned before, quantisation can adapt how the parameters of ANNs are represented. It is focused on optimising ANNs by reducing their size and their computational cost. Most of the available frameworks for training ANNs use 32 bits for parameter representation, but some micro-controllers are not able to make a correct interpretation of these networks unless they are quantised into eight bits.

### 2.4. Deep Learning in Edge Computing

Once the significance of both EC and DL has been contrasted separately, the next step is to check the current state of the combination of the two areas [45], focusing the analysis on the on-device intelligence architecture. The combination of these fields has been used in different areas such as image recognition, voice control systems, predictive maintenance for industry, health monitorisation, or even in information technology (IT) security.

Image analyses have been implemented in several applications, for example, a mobile application that is capable of identifying different meals [46]. This approach allows the control of the user’s diet by calculating the consumed calories. To do this, the user takes a photo of the meal using his/her mobile device, and the DL-based network performs a classification of the meal, computes the calories, and provides the required feedback. Other authors have presented different applications, such as [47], who presented an empty parking detection system, or [48], who presented an algorithm to predict the density of the people using public transport. Other authors have used image processing applied to sequential images such as video processing, offering the capability of studying the movement of objects in a video [49].

Object recognition systems have been analysed further with the objective of optimising DL networks, especially to reduce their latency. In this case, hybrid DL methods where tested on embedded systems [50]. They proved that combining algorithms such as Haar cascade [51] and CNNs led to systems with lower latency and similar accuracy. This is achieved by first applying Haar cascade on images to detect objects; then, these objects are cropped; and finally, a CNN performs the classification of these cropped objects.

As for voice control, there are potential commercial solutions such as Alexa and Siri. These systems do not carry out the voice recognition in the node, but they make a first processing of the signal, both for privacy issues and to avoid an excess of traffic in the network [52].

Systems have also been proposed for health monitoring, where sensors process the data obtained and then send them to a more centralised system where the user can visualise and understand the analysis [53].

All of these applications demonstrate that integrating DL into embedded systems can obtain excellent results. Even so, it can be seen that in most cases, these devices do not obtain any resource restriction, especially in cases where images or audio are being used. The most common devices in which DL is integrated are mobile phones or micro-controllers with a high computing and memory capacity.

Considering the presented approaches, DL algorithms can be applied in different situations, but their integration into embedded systems is still an open challenge, mainly due to the memory and computational restrictions. For this reason, the scope of this paper is to present a generalisation approach to using a DL algorithm in an EC device and to determine its real performance compared to commercial solutions and powerful machines.

### 2.5. Related Work

Recent works have shown interest in optimising the integration of DL algorithms into restrictive micro-controllers. In this aspect, two different approaches are currently carried out.


The first approach is focused on the design of DL networks. As stated in Section 2.3.3, there are several approaches to designing networks, but they are mostly applied in mobile devices or micro-controllers with high computational capabilities. Even though, works such as [54] have proven that it is possible to optimise them further, considering not only the total number of parameters and operations of the network, but also the low computational capabilities and the need for the low latency of these devices.


The second approach is more related to the methodology involved, both in the design and the integration processes. The authors of [55] proposed the TinyNAS method, which is a two-stage neural architecture search that includes parameters such as the specific constraints of the device, so that the search is not focused on finding the smallest networks, but is rather looking for the best network architecture that fulfils the specific constraint of the device. The paper [55] also introduced a memory-efficient inference library called TinyEngine. This library uses only the operations that are necessary to run the networks found with TinyNAS, which allows making full use of the limited resources in the micro-controller. Even though, this framework is focused only on integrating some specific networks, and so does not fulfil the objective of generalisation.


There are more frameworks in the literature that also provide capabilities to run DL networks in micro-controllers. For example, STM32Cube.AI (https://www.st.com/content/st_com/en/stm32-ann.html (accessed on 1 February 2021)) is able to execute DL networks trained in Keras, TensorFlow Lite, Caffe, ConvNetJs, and Lasagne, but it is a proprietary framework that can only be used in STM32 Arm Cortex-M-based micro-controllers.
Apache TVM [56] is another example; in this case, it is open-source, which means that it is also able to compile and run DL networks trained in TensorFlow, Caffe, or PyTorch, but it is mostly focused on integration into mobile devices (with Android or iOS operative systems) or any device with a Linux-based operating system. Apache TVM has also been working on a micro-controller-based library (microTVM), but it has only been tested in two different platforms.


Finally, Tensorflow Lite Micro [57] is another framework that integrates DL networks into micro-controllers. It implements most of the operations of CNNs, which makes it adequate for a generalist integration. However, the library by itself cannot control the device, as it does not provide any drivers, which is why it is compulsory to integrate it into another environment such as Mbed OS.


## 3. Methodology

### 3.1. Parameter Reduction

The optimisation of DL algorithms is crucial in order to integrate them with embedded systems, as they usually have limited resources. As explained in Section 2, one of the techniques that can be applied is the reduction of the size of DL networks, decreasing the total number of parameters or the way in which they are represented; this technique is parameter pruning and quantisation (PPQ) [29].

For the presented experiments, the quantisation technique was used to ensure that the resulting networks would be adequate for integration in any embedded system. The quantisation technique reduces the number of bits used to represent each parameter, allowing the integration of a full DL network into even a micro-controller with only an 8 bit architecture.

This technique changes the way the weights and activation of the DL network are represented. Each weight and activation are represented by a real number of 32 bits. When the network is quantised, the 32 bit representation is reduced to a smaller number, such as 16 or 8 bits. Generally speaking, quantisation is carried out with eight bits, because certain embedded systems are only capable of representing numbers in this format. However, it is possible to find quantisations of 16 bits or even 2 bits in extreme cases. This reduction leads to a loss of precision in the performance of the DL network, although there are several techniques that try to reduce this precision loss, e.g., retraining the network to adjust the new parameters.

Figure 3 (extracted from (https://blog.tensorflow.org/2020/04/quantization-aware-training-with-tensorflow-model-optimization-toolkit.html (accessed on 1 February 2021))) shows one of the techniques used for the experiment. The original range of the values of the weights and activation of the DL network were rescaled in order to fit in the maximum range of an 8 bit value ([−127,127]). The rescaling process was completed by dividing the original range in $2^{8}$ equidistant values and rounding the original values to the closest one. In this process, the real zero value is also taken into account for arithmetic optimisation (for further details, consult [58], Section 2.3).

Equation (Equation 2) represents the quantisation process, where *r* represents the real values, *S* represents the scale factor, *q* represents the quantised values, and *Z* represents the “zero-point”.
(2)r=S(q−Z)

In the specific case of quantisation carried out with TensorFlow, a symmetrical range is selected for the weights, so that the zero point will be 0, and an asymmetrical range is selected for the activation; thus, it is necessary to determine at which point the zero value is.

In order to quantise the parameters of the network, the first step is to determine the range [a,b] of the original values of the set of parameters. This range must fulfil the condition stated in Equation (Equation 3) that, with a scaling factor *S*, the zero value must be represented after quantisation. After defining the range, the quantised values are calculated as represented in Equation (Equation 4), where *n* refers to the number of quantisation levels (e.g., $2^{8}$ for 8 bit quantisation) and ⌊·⌉ denotes rounding to the nearest integer.
(3)$$a+Sx=0\;|x\in N\;\mathrm{and}\;x\leq \frac{b+a}{S}$$
(4)$$q\left(r;a,b,n\right)=\left\lfloor \frac{clamp(r;a,b)-a}{s(a,b,n)}\right\rceil s\left(a,b,n\right)+a$$

In this case, quantisation using ranges was used, since this technique has already been integrated into TensorFlow. Furthermore, quantisation can be applied in two different ways: (1) quantisation awareness training (QAT), and (2) after training quantisation (ATQ).

In QAT, the training itself takes into account the quantisation process. To do this, the DL network is first trained until it reaches the point of convergence. At this point, the network is quantised so that it loses precision, but to reduce this loss, the network is retrained for several steps ahead, considering the maximum bits that are allowed to represent the weights and activations. This allows readjusting the quantised parameters by varying the range of values just enough to increase the total precision of the network.

In ATQ, the network is quantised after its training has been completed, achieving lower accuracy results compared to QAT. In this case, two different techniques can be used: (1) quantise the weight matrix of the neural network directly (this is the simplest technique in which the the activation ranges are not directly quantised, so it cannot be considered that the network is totally quantised at 8 bits); (2) quantise the network weights and activations (in this case, the trigger ranges are indeed quantised). In the second case, it is necessary to have a representative data set (a data set similar or equal to the training set); the weight matrix is quantised in the same way as in the first option, and then, brief training is carried out focusing only on calibrating the activation ranges of the network so that the zero threshold is adjusted to the activation function. After this process, the network is fully quantised, so it can be integrated inside any embedded system. However, as the weights of the network are not readjusted, the accuracy is generally lower than that of QAT.

### 3.2. Environment

One of the objectives of this paper, besides the integration of ML algorithms into embedded systems, was that this process should be generalised for any architecture. The most important micro-controller manufacturers already have their own tools that facilitate this task, but the generalisation proposed in this manuscript tries to present a homogeneous integration process for any kind of embedded system.

Figure 4 shows the structure of the environment used, where the TensorFlow Lite for Micro-controllers (https://www.tensorflow.org/lite/microcontrollers (accessed on 1 February 2021)) library is integrated into the Mbed OS-based project. Mbed OS is used as a minimalist open-source operating system that groups together several drivers for embedded systems of major manufacturers, such as STMicroelectronics (https://www.st.com/content/st_com/en.html (accessed on 1 February 2021)) or NXP Semiconductors (https://www.nxp.com/ (accessed on 1 February 2021)). As all of the drivers necessary to control the micro-controllers are included in a single minimalist operative system, and it facilitates the development of applications for different devices. The limitation of this library is that it can only handle devices with an ARM processor. In addition, TensorFlow Lite for Micro-controllers is included in Mbed OS, so that, apart from the common drivers of the devices, the developer can also use the interpreter of this library to execute DL networks on the micro-controller. TensorFlow Lite for Micro-controllers is a library, currently under development, that allows the execution of DL networks trained in TensorFlow in micro-controllers. Table 1 summarises the different software used in the environment.


TensorFlow Lite for Micro-controllers is an extension included in the main repository of TensorFlow GitHub (https://github.com/tensorflow/tensorflow/tree/master/tensorflow/lite/micro (accessed on 1 February 2021)). This library requires the direct intervention of the user, because is does not work out of the box by default. In this regard, it is necessary to include minor corrections in the source code to allow integration with Mbed OS.

#### Integration Process

Figure 5 depicts the pipeline followed to allow a DL network architecture to fit correctly into an EC device. The first stage is the network preparation; to do so, a DL network must be developed and trained following the explanations given by the TensorFlow library (an example with details can be found at https://github.com/tensorflow/tensorflow/blob/master/tensorflow/lite/micro/examples/person_detection/training_a_model.md (accessed on 1 February 2021)). This training process should be accomplished by using a high-resource device (as explained in the Introduction of this paper). At this point, there are two different approaches to follow: (1) use the quantisation algorithms to modify the training process and to produce a final model with its weights quantised; (2) use the fully trained network (in 32 bits) and apply a quantisation process to reduce its weights. This process can be achieved easily using TensorFlow Lite Converter (details can be found at https://www.tensorflow.org/lite/convert (accessed on 1 February 2021)). Once the final network has been compressed, the next step is to convert the used Python code into a C-type array to allow the EC devices to read the code. This can be achieved by using the “xdd” GNU/Linux command, which creates hexdumps of a given file or a standard import (https://www.tutorialspoint.com/unix_commands/xxd.htm (accessed on 1 February 2021)).


Once these steps have been finished, the code is finally ready to be integrated into any Mbed OS-based project. To do so, it is necessary to create an Mbed OS project and link it with the required libraries to execute the code.


### 3.3. MobileNet

Once the development environment has been configured, it is necessary to select and train the required DL network to fulfil the requirements of reducing its size to fit in an EC device. To do that, we propose the use of an application that is capable of detecting the presence of a person based on an image captured by a camera. Thus, our main intention was to be able to fit a computer vision-based DL network into an EC device and to analyse its performance.

As far as selection is concerned, the reasons were focused exclusively on the size, as the platforms that were used have memory restrictions. Moreover, considering that the aim was to generalise the process of integrating DL networks into embedded systems, selecting the smallest network allows determining the minimum amount of resources required for image classification.

Table 2 (extracted from https://www.tensorflow.org/lite/guide/hosted_models (accessed on 1 February 2021)) shows a comparison of different DL networks dedicated to image classification. The model size refers to the size of the weight matrix of each neural network; the accuracy is the percentage of success of the network; and the latency is the time it takes to run the network. In all networks, MobileNet V1 [7] has the smallest weight matrix, indicating that it is the network that needs the smallest memory space. Besides, both the MobileNet V1 and the MobileNet V2 [40] networks have a parameter (depth multiplier) that allows them to be reduced in size, which affords them a certain flexibility to be integrated into systems with few resources.

Considering that the embedded system where the network is integrated has barely 2 MB of memory, the reduced MobileNet V1 seems to be the only valid option, because the rest of the networks are too big to fit in it. Even so, MobileNet V2 and V3 were tested to verify if it is possible to compress them to the point that they can be integrated into an embedded system using the explained techniques such as quantisation (see Section 3.1).

We selected versions V1, V2, and V3 of MobileNet as the DL architectures to be used in the experiments presented in this paper, as they allow the application of compression techniques in a reasonable way, providing the opportunity to focus the study on their accuracy and latency in an EC device.

#### 3.3.1. MobileNet V1

MobileNet V1 [7] was introduced in 2017 by Google. They highlighted that this network allows the adjustment of the memory and latency restrictions of the application to be implemented because it is designed to scale the size of each of its layers by modifying a depth factor.

Table 3 shows the architecture of the original design of MobileNet V1, which is based on the concatenation of standard convolutional filters (conv) and depthwise separable convolutional filters (conv dw) (explained in [7], Section 3.1). Each of these layers also includes a batch normalisation (BN) [59] and a ReLU [11] neural activation, as can be seen in Figure 6. The difference between standard and separable convolution layers is that separable ones are applied independently to each channel of the image, which decreases the latency of the network. Finally, average pooling (Avg Pool) reduces the spatial dimension to 1, and a fully connected layer (FC) with Softmax activation performs the final classification.

One of the highlights of this network is that, as mentioned before, the designers proposed a parameter to reduce the size of the network. This is done by multiplying a factor (depth multiplier) by the size of all the convolutional filters; thus, it decreases the size of the network uniformly. In addition, the designers also considered the reduction of the input images so that the latency decreases.

#### 3.3.2. MobileNet V2

In 2019, the same Google team published MobileNet V2 [40], where they modified the structure of the depthwise separable convolutional filters to bottleneck depth-separable convolutional (bottleneck filters). Figure 7 shows the structure, where the main objective was to improve the efficiency of memory consumption with respect to the filters presented in V1. Taking into account the improvement of these blocks, the design was reworked, defining the architecture shown in Table 4.

#### 3.3.3. MobileNet V3

MobileNet V3 [41] was also published in 2019. In this case, the filters of the previous versions were combined with a block called squeeze and excite, introduced in [60]. Figure 8 (extracted from https://towardsdatascience.com/review-senet-squeeze-and-excitation-network-winner-of-ilsvrc-2017-image-classification-a887b98b2883 (accessed on 1 February 2021)) represents how this block is applied to an inception [61] neural network architecture, but this concept can be extrapolated to the filters proposed for V1 and V2.

In V3, they used an automatic algorithm to search for DL network structures [62] to define a base structure, which they made small adjustments to so as to optimise it further. In this case, the network does not consider a depth multiplier parameter as its predecessors did. Instead, two different networks were designed during its conception, one being V3-large and the other V3-small.

The difference between these two networks is that V3-small has 4 fewer bottleneck layers than V3-large; thus, its size is reduced. In this case, as the platform has memory restrictions, MobileNet V3-small was used, whose architecture is available in Table 5.

Finally, as explained in Section 3.2, TensorFlow was used to train all of these networks (https://github.com/tensorflow/models/tree/master/research/slim/nets (accessed on 1 February 2021)). Table 6 depicts the used training parameters. These training parameters were used in all network configurations except quantisation delay, which was only used in QAT networks, indicating in which step it was applied.

Table 7 depicts the used training environment specifications to train the presented MobileNet algorithms with their different input parameters. Considering that MobileNet is a computer vision DL network, the used training environment should be a robust machine to train the algorithms in a reasonable time. With the presented configuration, the training time took four days.

### 3.4. Experimental Platforms

Three different platforms were used to carry out the experiments: (1) OpenMV Cam H7 (https://openmv.io/products/openmv-cam-h7 (accessed on 1 February 2021)), which is a platform specifically designed to implement AI-based applications for image processing; we considered the usage of this device because it can be used as a baseline to compare the performance of the generalist development environment implemented with a specifically designed environment; (2) STM32H747i-Disco (https://www.st.com/en/evaluation-tools/stm32h747i-disco.html (accessed on 1 February 2021)), as a platform to integrate the developed environment and to analyse the performance; (3) finally, a desktop computer, to compare the performance of the DL algorithms integrated into a memory-limited environment or without integration to check if the proposed algorithms are capable of running faster in an EC device or in a desktop computer.

#### 3.4.1. OpenMV Cam H7

OpenMV Cam H7 is a device specifically designed to implement applications for image processing using AI. Table 8 shows the system’s technical specifications. The device does not have an OS installed by default; instead, it has customised firmware that includes a Python programming language interpreter and a TensorFlow Lite for Micro-controllers library, which facilitates a rapid prototyping of multiple AI applications. Table 9 shows the used version of the OpenMV firmware, as well as the version of the library used for image processing (Pillow).

The main limitation of this device is that the integration of the TensorFlow library is carried out in such a way that it only accepts working with an 8 bit architecture, which forces quantising the DL networks. In addition, it does not let the researchers customise their software to create generalised solutions, limiting the creation of different potential solutions to test the quantisation process in the device itself.

#### 3.4.2. STM32H747i-Disco

STM32H747i-Disco is an evaluation board that is often used for the prototyping of different applications. In the embedded system’s field, it can be considered quite a powerful system, as it has two processors and a wide range of peripherals (the features are available in Table 10).

However, the STM manufacturer has a development tool called ST-X-CUBE-AI (https://www.st.com/en/embedded-software/x-cube-ai.html (accessed on 1 February 2021)) that allows integrating DL algorithms into STM micro-controllers. This tool supports DL networks trained with different libraries, such as TensorFlow or Caffe, and optimises the usage of memory automatically. The problem lies in the fact that this development is tied to the manufacturer itself and its licenses, so it is not aligned with the objective of generalising the process.

With this assumption, the experiments were focused on using STM32H717i-Disco with the presented environment (detailed in Section 3.2). Furthermore, as the aim of this research was to test the integration of DL networks in memory-restricted embedded systems, the capacity of the board was limited to the exclusive use of the ARM Cortex M7 CPU and the memory usage to only 2 MB of flash and 1 MB of RAM (restricting the usage of SDRAM and SPI NOR flash memory). This limitation of the device’s capacity also helped to make a fair comparison with the OpenMV Cam H7 system, since, in this way, both devices had the same specifications.

As for the software used, the development environment used is that explained in Section 3.2, where different libraries were also used to transform DL networks and images into C-type arrays. Table 11 shows the libraries and versions used.

#### 3.4.3. Personal Computer

Finally, a PC was used to ensure that the performance of the DL network did not decay compared to its performance in a non-limited system such as a computer. The characteristics of the computer used are available in Table 12. In order to test the neural network in the computer, the library TensorFlow Lite (https://www.tensorflow.org/lite (accessed on 1 February 2021)) was used. This library is the basis of the TensorFlow Lite for Micro-controllers library, but it has more functionalities implemented. Table 13 shows the versions of all of the libraries involved in the process.

### 3.5. Metrics

The experiments consisted of comparing the functioning of a neural network for binary image classification (the image contains a person vs. the image does not contain a person); thus, the selected metrics had to represent their performance. The usual metric used in these cases is accuracy (Equation (Equation 5)). The accuracy represents the total percentage of right guesses based on the whole set. This metric is useful to see the overall performance of the network when the data set is balanced, but it is not able to determine in which cases it fails. For that, the precision, recall, and F1 metrics were also evaluated.

Precision focuses on showing the success rate only for predictions where it is determined that there is a person in the picture (Equation (Equation 6)). Recall focuses on showing the success rate only for those cases where there are actually people in the picture (Equation (Equation 7)). Finally, F1 is a combination of precision and recall in the form of a harmonic mean (Equation (Equation 8)), which manages to show a complete summary of the confusion matrix.

In addition to these metrics, it is also interesting to compare the performance of the integration of the DL network into the embedded system. The most critical point is the network inference time, i.e., how long it takes to classify an image. This metric was only used in the experiments where the platforms had the same characteristics, since it is not considered a fair comparison for experiments where one platform has a higher computational capacity than the other.
(5)$$accuracy=\frac{TP+TN}{TP+TN+FP+FN}$$
(6)$$precision=\frac{TP}{TP+FP}$$
(7)$$recall=\frac{TP}{TP+FN}$$
(8)$$F_{1}=2\times \frac{precision\times recall}{precision+recall}$$

### 3.6. Data Set

Visual wake words (VWW) [63] is a set of images specifically designed to train compact DL networks with the aim of integrating them into micro-controllers. VWW is derived from a more complex data set called common objects in context (COCO) (https://cocodataset.org/ (accessed on 1 February 2021)). COCO contains 123,287 images for object detection, where in each of them, the objects (up to 80 different ones) and their contours in the image are labelled.

VWW simplifies the COCO data set, transforming it into a binary classification to predict whether there is a person in the picture. To do this, it edits the original COCO labels, restricting them into “person” or “non-person”. This decision is made based on whether the original image already has an object “person” in it and whether the area of that object is larger than 5% of the entire image area. With this simple algorithm, all COCO tags are transformed into binary tags. VWW can be obtained via a script available from TensorFlow (https://github.com/tensorflow/models/blob/master/research/slim/datasets/download_and_convert_visualwakewords.py (accessed on 1 February 2021)).

The data set is already divided so that approximately 8000 images are used for testing and the remaining 115,287 for training. For experimentation purposes, the training set was retained, but the test set was drastically limited to a total of 100 images, where 50 of them contained people and 50 of them did not. This comes from the limitation of the memory of the embedded system used, where it was impossible to include the whole set; therefore, it was decided to reduce it. Figure 9a,b shows some of the images used for the experimental phase.

Furthermore, these images were transformed to match the input tensor defined for each of the networks to be tested. In this case, the images were resized to the resolution used by each network and converted to grey-scale.

These transformations were necessary due to the limitation of the device’s memory, since the higher the resolution of the image, the larger the input tensor was; therefore, it increased the RAM consumption. With the grey-scale transformation, the same thing happened, since by reducing the channels of the image (from red-green-blue (RGB) to grey-scale), the input tensor also reduced its dimension.

## 4. Experiments

In this section, we show the results of the experiments exposed in the previous section. The overall idea is to show our results after conducting different training and testing in the presented experimental platforms. Then, in the next section, we discuss the obtained results, offering different reasons for why we obtained said results.

### 4.1. Experimental Phase 1

In the first experimental phase, the aim was to ensure that the proposed development environment can integrate DL networks into embedded systems. MobileNet V1 and V2 were trained and their performance was tested on both STM32H747I-Disco and OpenMV Cam H7.

Table 14 depicts the experimental process followed. The ID column refers to the experiment name; the Platform column refers to the platform used; the Network column shows the network used in the tests; the Depth Multiplier column refers to the compression factor in which the layers of the neural networks were reduced; the Resolution column refers to the size of the images used for testing; finally, the Quantisation column indicates the applied quantisation process in each network. Note that only the ATQ was applied, using a representative data set to also quantise the neural network activations (the explanation is given in Section 3.1).

### 4.2. Experimental Phase 2

In the second experimental phase, the aim was to ensure that the performance of the DL networks did not decay when they were integrated into an embedded system (i.e., an EC device). In order to verify that the embedded system is capable of executing DL networks, different networks were trained with different sizes, complexities, and input sizes. All of them were tested in both STM32H747I-Disco and the PC experimental platform.

The size of the networks was reduced using the depth multiplier parameter. In addition, the input tensor was changed for some networks to check whether increasing the resolution of the images could improve the performance, despite the increase of RAM consumption. Table 15 shows the proposed experiments, whose characteristics maintained the same structure as the experiments of the previous phase.

### 4.3. Experimental Phase 3

In the third and final phase, the aim was to analyse the effect that the quantisation process had on the neural network. In this case, MobileNet V2 was used as a reference, and it was trained by applying quantisation during and after the training. The performance of both networks was then tested on the STM32H747I-Disco platform. In addition, the compression ratio of the network after applying different quantisation techniques was also verified. Table 16 shows the proposed experiments, which also follow the same structure as the previous experiments.

## 5. Results and Discussion

### 5.1. Experimental Phase 1

In order to verify how the proposed method worked in the development environment, a comparison of the obtained predictions given by the outputs of each DL network was conducted to ensure that they were identical in both environments. Table 17 shows the results of the execution of two DL networks in two different experimental platforms. The results of MobileNet V1 in the E1-STM-V1and E1-OMV-V1experiments showed that this network did not perform adequately for the problem we were trying to solve, as it could not distinguish people in any image. This happened because the original structure of the network was designed to be as simple as possible, and the size of the convolutional layers was reduced even further to 25% of the original design, loosing the object recognition capability.

Table 17 shows how OpenMV Cam H7 had a lower latency, which indicates that its environment was better optimised than the others. The difference was almost 50 ms of execution time for each processed image, which might be critical depending on the scenario. Taking into account that the environment explained in Section 3.2 tries to generalise the integration of DL networks into embedded systems, the obtained results are promising against a system that was specifically designed for the purpose of executing DL algorithms.

Table 17 also depicts a comparison of the behaviour of MobileNet V1 and V2. As it is possible to see in the table, MobileNet V2 was capable of identifying people in black and white images of 96×96 pixels with an 88% success rate in the test set, despite having reduced its convolutional layers to 10% of the size of the original design. This shows that the development environment proposed in this paper is capable of classifying images with acceptable accuracy using an embedded system. In addition, the fact that it was not possible to run MobileNet V2 on the OpenMV Cam H7 platform demonstrates the benefits of having a generalised development environment. In some embedded systems where the firmware is given, the manufacturers tend to wait for a major version of the used technical libraries before they release a new version of their firmware. This can be a disadvantage if the developer is working with recent versions of some specific libraries that are under heavy development. Therefore, having the option to update the libraries in a flexible way is an important feature for a development environment.

### 5.2. Experimental Phase 2

Table 18 shows the obtained results in the second experimental phase using DL networks for integration into the STM32H747i-Disco platform. MobileNet V1 was not able to identify people, even if the provided input image size was increased to 128×128 pixels (E2-(STM&PC)-V1-025-128). In contrast, MobileNet V2 (E2-(STM&PC)-V2-010-96), whose weight matrix was similar in size to E2-(STM&PC)-V1-025-128, was capable of identifying people. This comes from the benefits offered by the bottleneck depth-separable convolutional filter compared to the depth-separable convolutional filter, which, as stated in Section 3.3.2, was designed to optimise memory consumption. The optimisation led to a more accurate DL network without increasing its size. This fact highlights the importance of having a network structure itself that, depending on how effective it is for a particular task, can maintain its performance, potentially even reducing the original size of the convolutional layers.

Table 19 shows the performance of the networks that were not properly integrated into the embedded system. For some networks (E2-(STM&PC)-V2-035-96 and E2-(STM&PC)-V3-96), we could not verify the RAM consumption, as it was necessary to carry out an initial integration of the network on the platform. In cases where the weight matrix was too large, the RAM consumption of these networks was not calculated (They are marked with an * in Table 19).

The E2-(STM&PC)-V1-025-224 experiment supports the hypothesis that the size of the image is not as crucial as the structure of the network itself. Even with 224×224 pixel images, MobileNet V1 still could not identify people, which may be surprising, considering that this network is larger than E2-(STM&PC)-V2-010-96 and consumes twice as much RAM. These results demonstrate that the MobileNet V1 network, after reducing the size of the filters and quantising it to eight bits, lost the ability to extract the features of an image; thus, the network does not work for image classification tasks, as it lost the most crucial benefit of CCNs.

Regarding experiments E2-(STM&PC)-V2-010-96, E2-(STM&PC)-V2-015-96, and E2-(STM&PC)-V2-035-96, it can be seen that the size of the convolutional layers of a particular structure (in this case, MobileNet V2) is decisive in terms of performance. A substantial improvement can be observed if the performance when the network that was decreased to 10% and the network that was decreased to 35% are compared, but this improvement came at the cost of a substantial increase in the size of the weight matrix. Therefore, for a real application, after selecting the most suitable DL network structure, a balance between the size and its performance should be found.

Figure 10 shows that there was no direct relationship between the size of the weight matrix and the F1 obtained by that network. It can be seen that, for example, the V2_015_96px network achieved similar results compared to a network that was five times bigger (V3_small). Moreover, the smallest network, V2_010_96px, outperformed the MobileNet V1 network, which was similar in size.

MobileNet V3 is, in this case, the biggest network, and even with a 92% accuracy, it was not the best. The design of this network was automated, and the enhancements it made were focused on reducing the latency and making it more quantisation friendly [41]. Due to this, its performance in low-resource devices was similar to that of MobileNet V2, but the latency should be lower. This fact was not proven, as the memory restrictions defined for the experimentation precluded the integration of the network, as it did not have the option to automatically reduce its size.

Figure 11 shows a similar comparison, but in this case referring to RAM consumption and F1. This figure also shows that there was no direct relationship between them, as the network with the highest consumption had one of the worst performances. It was already corroborated that it is possible to implement a DL network with less RAM consumption and to achieve better results if the structure and the filters used are better optimised.

After verifying that the performance of the network was not directly related to the size of the weight matrix, or to the RAM consumption, the next step was to analyse whether there was a relationship between these two characteristics. Figure 12 shows that, for both MobileNet V1 and V2, there was no relationship between RAM consumption and the size of the weight matrix. In the case of MobileNet V1, as the resolution of the images increased, the RAM consumption increased, but the size of the weight matrix did not. In the case of MobileNet V2, on the contrary, the network with larger convolutional layers consumed more RAM, but the size of the weight matrix did not increase as much. This means that the consumption of RAM and the size of the weight matrix were not directly related. After all, the RAM consumption was determined mostly from the biggest filter used in the network and not from the total number of parameters. Taking this into account, it is possible to manage the structure of the network, taking into account that big filters will increase RAM consumption and multiple small filters will increase the weight matrix.

### 5.3. Experimental Phase 3

Finally, the third experimental phase did not show any reliable results. Table 20 shows that QAT MobileNet V2 could not identify people in images. As stated in Section 5.3, there is a bug in the TensorFlow quantisation process (https://github.com/tensorflow/model-optimization/issues/368 (accessed on 1 February), https://github.com/tensorflow/models/issues/8816 (accessed on 1 February)). The implementation of one type of convolutional layer (DepthConv) had an error that activated the quantisation too early. This precluded the training of the network, and it only tried to adapt the quantised parameters. The bug has now been corrected by TensorFlow and will be included in the next official version. The correction is now available in the nightly version, where all changes are tested before an official version is released. Without this error, QAT should be able to improve the performance of ATQ, as it trains the network after being quantised, obtaining more accurate parameters.

### 5.4. Embedded Device Integration

Various operating systems are widely available for the Internet of Things (IoT) environment using embedded devices based on micro-controllers with the characteristics of being low-powered, constrained, and connected [64,65]. At the same time, different environments designed to train ML algorithms in desktop computers to make inferences in micro-controllers have also been tested in the literature [66]. This paper generalised an environment based on Mbed OS and TensorFlow Lite. Mbed OS is a free, open-source operating system for the IoT based on ARM micro-controllers, which represent the most common basic devices that can run DL algorithms. TensorFlow Lite Micro is an open source library that enables developers run TensorFlow DL models on embedded and IoT devices. Since most of the currently existing software development platforms for micro-controllers are proprietary (e.g., STM32Cube.AI (https://www.st.com/content/st_com/en/stm32-ann.html (accessed on 1 February))), the proposed environment has the characteristics of not depending on a specific manufacturer or proprietary software.


The generalisation of the previous environment is based on the availability that offers the usage of Mbed OS, which allows implementing applications for ARM-based embedded devices. Moreover, the TensorFlow Lite Micro library used for the execution of DL networks is a C/C++-based library, which can be integrated into any kind of device that has enough memory capacity. Finally, a quantisation [58] technique was used in this paper to reduce the size of different DL networks to adapt the computing needs of the network to the micro-controller resources. Both the QAT and ATQ mathematical techniques can be implemented in any software intended for machine learning analysis, since they are based on reducing storage memory needs at the cost of losing precision with the benefit of improving inference speed.


## 6. Conclusions

In this paper, a development environment for the integration of DL networks into embedded systems was proposed, and an analysis of its performance was conducted. First, the bases of both DL algorithms and EC devices were studied, analysing the current state-of-the-art of both, as well as their existing applications. Together with the study of these fields, the quantisation technique used for the experiments was detailed.

Furthermore, a development environment was proposed, which allows integrating DL networks into a wide variety of embedded systems with a homogeneous process. This environment is based on combining the Mbed OS operating system with the TensorFlow Lite for Micro-controllers library, which allows executing DL networks in micro-controllers not designed for this purpose.

As for the experiments carried out, three phases were proposed. The first experimental phase focused on comparing the functioning of DL networks in a system designed for this purpose and in the environment proposed. The second experimental phase tried to verify that the proposed development can correctly execute DL networks, comparing the integration with the results obtained using a PC. For this comparison, several networks with different parameters were trained so that their behaviour could be analysed. The third and last experimental phase focused on comparing QAT with ATQ.

The first phase of the experiments verified that the development environment proposed in Section 3.2 can execute DL algorithms and that, despite the fact that OpenMV Cam H7 executed the network faster, the generalisation of the development environment allowed adapting or updating the libraries, allowing executing more complex networks, which OpenMV Cam H7 was not able to do. This confirms the value that a generalist development environment can provide. The second phase verified that the results obtained in the proposed environment were identical to the results obtained using a computer. In addition, the network performance was analysed, comparing the image resolution, the size of the weight matrix, and the RAM consumption. In this respect, increasing the resolution of the images did not improve the performance of MobileNet V1, despite its size increase. Furthermore, larger networks or networks with higher RAM consumption did not necessarily achieve better results, but if the same network structure was maintained, increasing the size of the layers indeed improved performance, although the system needed more memory resources.

Finally, it was confirmed that for the binary classification of images, a highly compressed neural network can have a place in real applications, but the extrapolation of this possibility to more complex scenarios is not clear. Therefore, it is necessary to analyse the minimum resources required for more complex application environments such as object tracking. In addition, further work on optimising the use of neural networks in embedded systems should be considered. In this section, we distinguished at least three lines of research. Techniques for reducing the size of networks, such as quantisation or pruning, should continue to be improved so that networks do not lose their effectiveness after the training process. It is also possible to optimise the use of the resources of embedded systems in order to reduce latency and energy consumption, or even to be able to integrate larger networks by adjusting the frameworks that are used to the specific device. Finally, the integration could also be improved by designing the hardware of embedded systems meant to execute DL networks. This could lead to, for example, including GPUs for applications in computer vision.

## Figures and Tables

**Figure 1 sensors-21-01031-f001:**
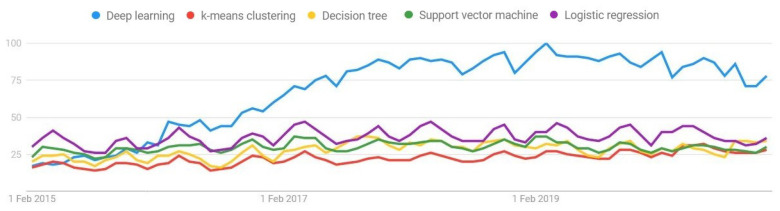
Google search trends of artificial intelligence (AI) algorithms.

**Figure 2 sensors-21-01031-f002:**
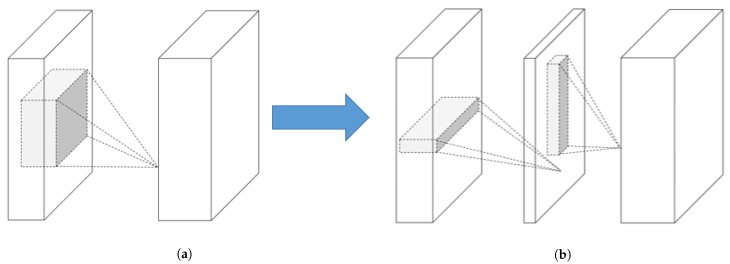
Decomposing the filter located on the left into two smaller filters (shown on the right). (**a**) Applying an *n* × *m* filter to an image; (**b**) Decomposing an *n* × *m* filter into *n* × *r* and *r* × *m* filters.

**Figure 3 sensors-21-01031-f003:**
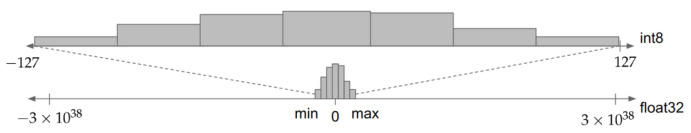
An 8 bit quantisation example.

**Figure 4 sensors-21-01031-f004:**
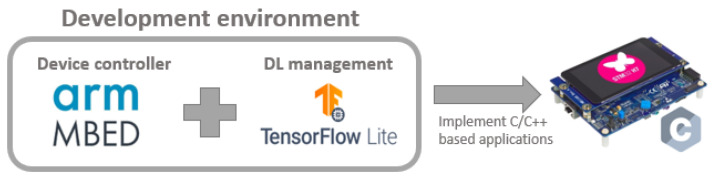
General structure of the development environment used for deep learning (DL) integration into embedded systems.

**Figure 5 sensors-21-01031-f005:**
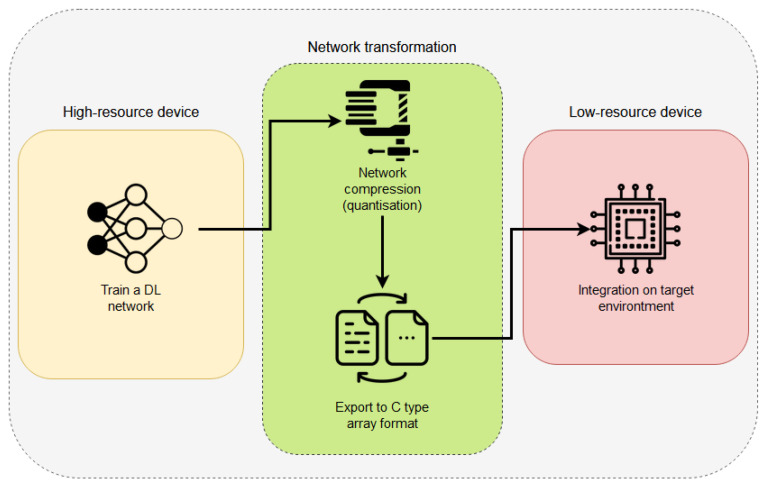
Pipeline followed to integrate a DL network architecture into an edge computing (EC) device.

**Figure 6 sensors-21-01031-f006:**
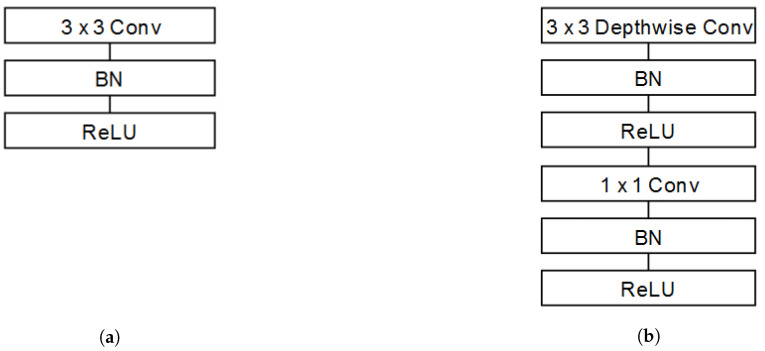
Differences among the structures of standard convolutional and depthwise separable convolutional filters (figures extracted from [7]). (**a**) Standard convolutional filter structure; (**b**) Depthwise separable convolutional filter structure.

**Figure 7 sensors-21-01031-f007:**
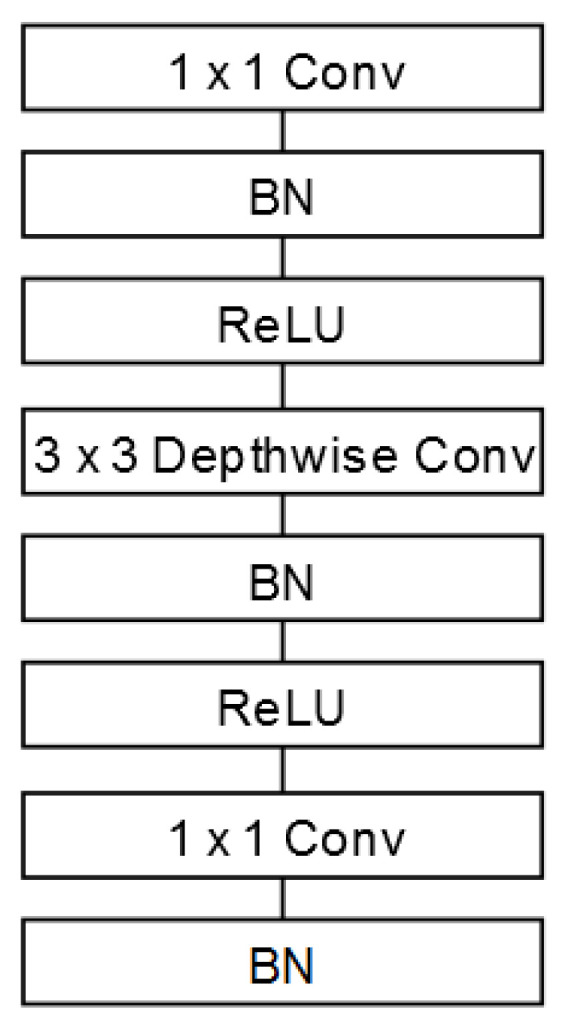
Bottleneck depth-separable convolutional filter structure.

**Figure 8 sensors-21-01031-f008:**
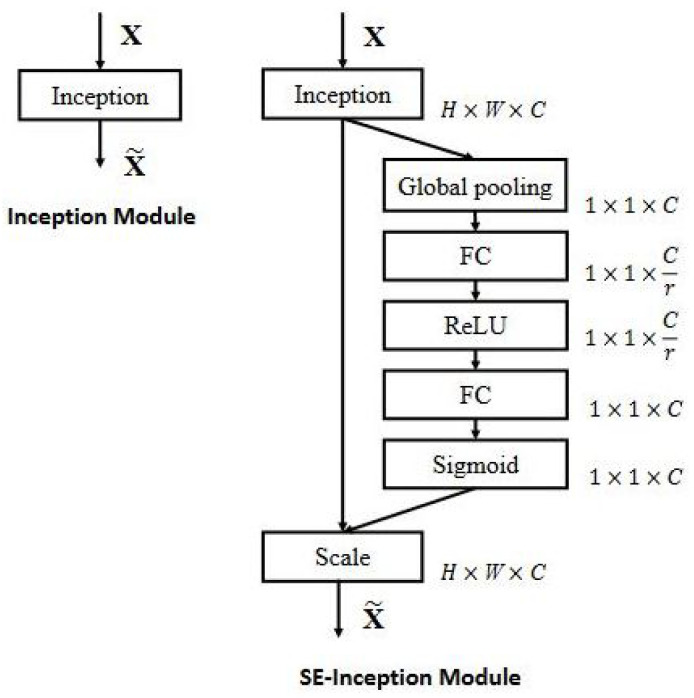
Example of squeeze and excite block integration in an inception neural network.

**Figure 9 sensors-21-01031-f009:**
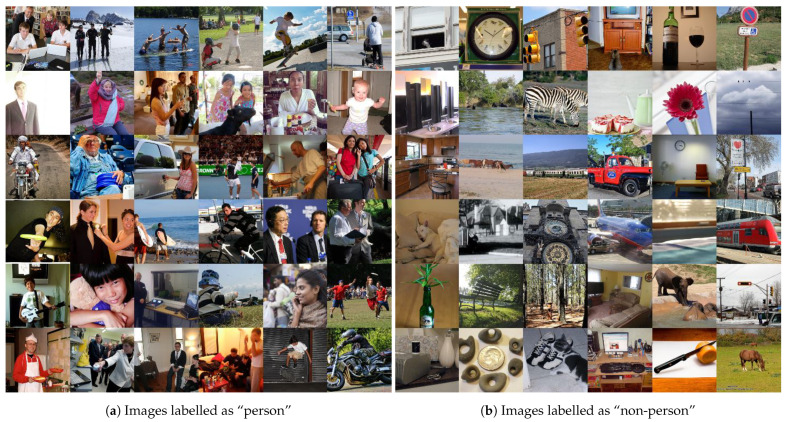
Sample of the images used for the experimental phase.

**Figure 10 sensors-21-01031-f010:**
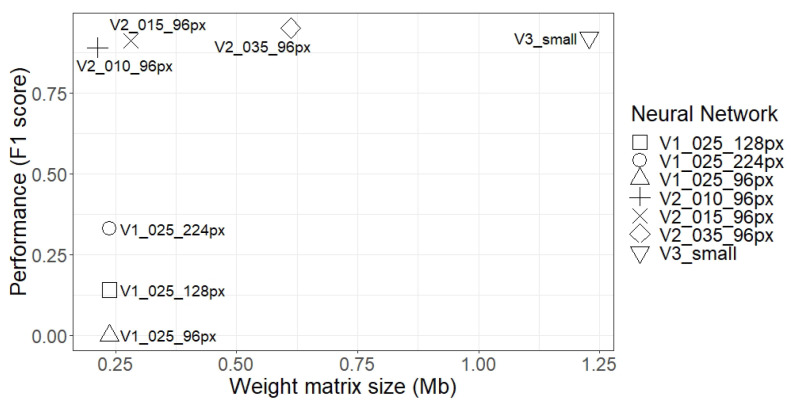
Comparison of the performance of the tested networks (F1) and the size of their weight matrices.

**Figure 11 sensors-21-01031-f011:**
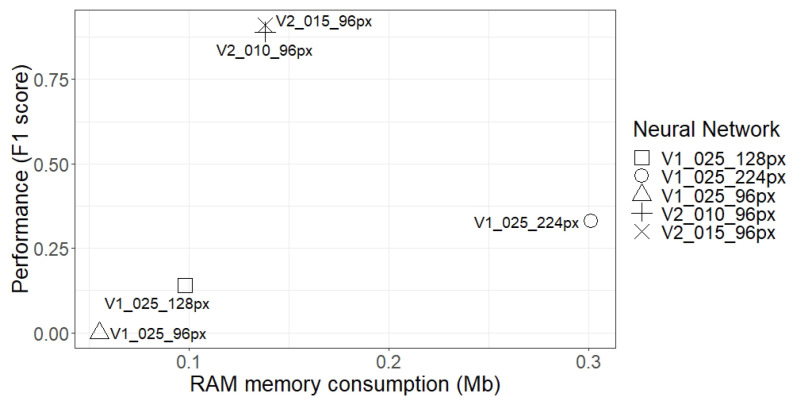
Comparison among the performance of the tested networks (F1) and their RAM consumptions.

**Figure 12 sensors-21-01031-f012:**
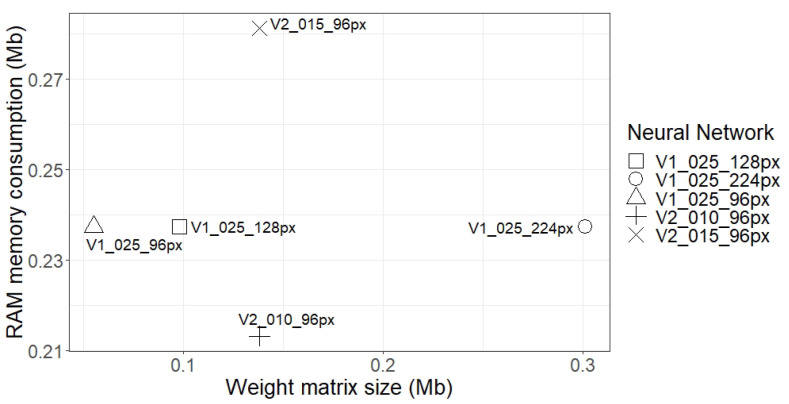
Comparison among the sizes of the weight matrices of the tested neural networks and their RAM consumptions.

**Table 1 sensors-21-01031-t001:** Software versions used for the environment.

Mbed OS	TensorFlow
5.15.3	GitHub source code (21 June 2020)

**Table 2 sensors-21-01031-t002:** Image classification DL network comparison.

Model Name	Size (MB)	Accuracy	Latency (ms)
DenseNet	43.6	0.642	195.0
SqueezeNet	5.0	0.490	36.0
NASNetmobile	21.4	0.739	56.0
NASNet large	355.3	0.826	1170.0
ResNet_V2_101	178.3	0.768	526.0
Inception_V3	95.3	0.779	249.0
Inception_V4	170.7	0.801	486.0
Inception_ResNet_V2	121.0	0.775	422.0
MobileNet_V1_0.25_128	1.9	0.414	1.2
MobileNet_V1_1.0_128	16.9	0.652	9.0
MobileNet_V2_1.0_224	14.0	0.718	17.5

**Table 3 sensors-21-01031-t003:** MobileNet V1 architecture. Type represents the type of filter used in the layer, and Stride represents the number of pixels by which the filter is shifted (table extracted from [7]). dw, depthwise.

Type/Stride		Filter Shape	Input Size
Conv/s2		3×3×3×32	224×224×3
Conv dw/s1		3×3×32 dw	112×112×32
Conv/s1		1×1×32×64	112×112×32
Conv dw/s2		3×3×64 dw	112×112×64
Conv/s1		1×1×64×128	56×56×64
Conv dw/s1		3×3×128 dw	56×56×128
Conv/s1		1×1×128×128	56×56×128
Conv dw/s2		3×3×128 dw	56×56×128
Conv/s1		1×1×128×256	28×28×128
Conv dw/s1		3×3×256 dw	28×28×256
Conv/s1		1×1×256×256	28×28×256
Conv dw/s2		3×3×256 dw	28×28×256
Conv/s1		1×1×256×512	14×14×256
5×	Conv dw/s1	3×3×512 dw	14×14×512
	Conv/s1	1×1×512×512	14×14×512
Conv dw/s2		3×3×512 dw	14×14×512
Conv/s1		1×1×512×1024	7×7×512
Conv dw/s2		3×3×1024 dw	7×7×1024
Conv/s1		1×1×1024×1024	7×7×1024
Avg Pool/s1		Pool 7×7	7×7×1024
FC/s1		1024×1000	1×1×1024
Softmax/s1		Classifier	1×1×1000

**Table 4 sensors-21-01031-t004:** MobileNet V2 architecture: Each line describes one or more identical layers repeated *n* times. All layers in the same sequence have the same number *c* of output channels. The output of each layer is determined by applying the following equation: $\frac{w}{s}\times \frac{h}{s}\times \frac{\left(tk\right)}{s}$, where *w*, *h*, and *k* represent the input of the layer. *s* defines how many pixels are shifted in the filter, and *t* represents the expansion factor (table extracted from [40]).

Input	Operator	*t*	*c*	*n*	*s*
$224^{2}\times 3$	conv2d	-	32	1	2
$112^{2}\times 32$	bottleneck	1	16	1	1
$112^{2}\times 16$	bottleneck	6	24	2	2
$56^{2}\times 24$	bottleneck	6	32	3	2
$28^{2}\times 32$	bottleneck	6	64	4	2
$14^{2}\times 64$	bottleneck	6	96	3	1
$14^{2}\times 96$	bottleneck	6	160	3	2
$7^{2}\times 160$	bottleneck	6	320	1	1
$7^{2}\times 320$	conv2d 1×1	-	1280	1	1
$7^{2}\times 1280$	avgpool7×7	-	-	1	-
1×1×1280	conv2d 1×1	-	k	-	2

**Table 5 sensors-21-01031-t005:** Structure of the MobileNet V3-small architecture. *#out*refers to the output channel, SE denotes whether there is a squeeze and excite in that block, and NL denotes the type of nonlinearity used. Here, HS denotes h-swish, and RE denotes ReLU. Finally, NBN denotes that no BN is applied in the layer (table extracted from [41]).

Input	Operator	Exp Size	*#Out*	*SE*	NL	*s*
$224^{2}\times 3$	conv2d, 3×3	-	16	-	HS	2
$112^{2}\times 16$	bneck, 3×3	16	16	✓	RE	2
$56^{2}\times 16$	bneck, 3×3	72	24	-	RE	2
$28^{2}\times 24$	bneck, 3×3	88	24	-	RE	1
$28^{2}\times 24$	bneck, 5×5	96	40	✓	HS	2
$14^{2}\times 40$	bneck, 5×5	240	40	✓	HS	1
$14^{2}\times 40$	bneck, 5×5	240	40	✓	HS	1
$14^{2}\times 40$	bneck, 5×5	120	48	✓	HS	1
$14^{2}\times 48$	bneck, 5×5	144	48	✓	HS	1
$14^{2}\times 48$	bneck, 5×5	288	96	✓	HS	2
$7^{2}\times 96$	bneck, 5×5	576	96	✓	HS	1
$7^{2}\times 96$	bneck, 5×5	576	96	✓	HS	1
$7^{2}\times 96$	conv2d, 1×1	-	576	✓	HS	1
$7^{2}\times 576$	pool, 7×7	-	-	-	-	1
$1^{2}\times 576$	conv2d, 1×1, NBN	-	1024	-	HS	1
$1^{2}\times 1024$	conv2d, 1×1, NBN	-	k	-	-	1

**Table 6 sensors-21-01031-t006:** Training parameters.

Parameter	Description	Value
Learning rate	How much the network learns in each step	0.045
Label smoothing	Proportion to smooth the probabilities of the final labels	0.1
Number of epochs per decay	After how many epochs the learning rate is reduced	2.5
Learning rate decay factor	How much the learning rate is reduced	0.98
Moving average decay	Moving average decay applied to the weights	0.9999
Batch size	Size of the batches generated from the data set	96
Max number of steps	Total number of steps done in the training	1,000,000
Quantisation delay	Quantisation process will start after this number of steps	850,000

**Table 7 sensors-21-01031-t007:** Training server characteristics.

Component	Specification
Processor	Intel Xeon Gold 5118
Graphic card	NVIDIA Tesla V100 16 Gb
RAM	500 Gb
Hard drive	200 Gb

**Table 8 sensors-21-01031-t008:** Characteristics of the OpenMV Cam H7 platform.

Component	Specification
CPU	ARM Cortex M7
CPU speed	480 MHz
RAM	1 MB
Flash	2 MB
**Peripherals**	Micro SD
	Camera

**Table 9 sensors-21-01031-t009:** Software versions used by OpenMV Cam H7.

OpenMV	Pillow
3.6.2	6.2.2

**Table 10 sensors-21-01031-t010:** Characteristics of the STM32H747I-Disco platform.

Component	Specification
CPU	ARM Cortex M4+
	ARM Cortex M7
CPU speed	240 MHz (M4)+
	480 MHz (M7)
RAM	1 MB
Flash	2 MB
SDRAM	256 MB
SPI NOR flash	2×512 MB
Peripherals	Micro SD
	Audio In/Out
	Microphone
	USB
	Ethernet
	Touch screen
	Expansion board connectors
	Buttons + LEDs

**Table 11 sensors-21-01031-t011:** Software versions used by STM32H747I-Disco.

Mbed OS	TensorFlow	Pillow
5.15.3	GitHub source code (21 June 2020)	6.2.2

**Table 12 sensors-21-01031-t012:** Characteristics of the PC.

OS	Processor	RAM
Ubuntu 18.04.04	i7-8650U	16 Gb

**Table 13 sensors-21-01031-t013:** Software versions used by the PC.

Python	TensorFlow	Pillow
2.7.17	1.15.0	6.2.2

**Table 14 sensors-21-01031-t014:** Phase 1 experiments. ATQ, after training quantisation.

ID	Platform	Network	Depth Multiplier	Resolution	Quantisation
E1-STM-V1	STM	MobileNet V1	0.25	96×96	ATQ
E1-OMV-V1	OpenMV	MobileNet V1	0.25	96×96	ATQ
E1-STM-V2	STM	MobileNet V2	0.10	96×96	ATQ
E1-OMV-V2	OpenMV	MobileNet V2	0.10	96×96	ATQ

**Table 15 sensors-21-01031-t015:** Phase 2 experiments.

ID	Platform	Network	Depth Multiplier	Resolution	Quantisation
E2-(STM&PC)-V1-025-96	STM and PC	MobileNet V1	0.25	96×96	ATQ
E2-(STM&PC)-V1-025-128	STM and PC	MobileNet V1	0.25	128×128	ATQ
E2-(STM&PC)-V1-025-224	STM and PC	MobileNet V1	0.25	224×224	ATQ
E2-(STM&PC)-V2-010-96	STM and PC	MobileNet V2	0.1	96×96	ATQ
E2-(STM&PC)-V2-015-96	STM and PC	MobileNet V2	0.15	96×96	ATQ
E2-(STM&PC)-V2-035-96	STM and PC	MobileNet V2	0.35	96×96	ATQ
E2-(STM&PC)-V3-96	STM and PC	MobileNet V3	-	96×96	ATQ

**Table 16 sensors-21-01031-t016:** Phase 3 experimentation. QAT, quantisation awareness training.

ID	Platform	Network	Depth Multiplier	Resolution	Quantisation
E3-V2-QAT	STM	MobileNet V2	1	96×96	QAT
E3-V2-ATQ	STM	MobileNet V2	1	96×96	ATQ

**Table 17 sensors-21-01031-t017:** Results of the execution of two DL networks in two different experimental platforms (MobileNet V2 results running on the OpenMV Cam H7 platform are not available). The best experiment and the best results for each metric are marked in bold.

ID	Accuracy	Precision	Recall	F1	Latency (ms)
E1-STM-V1	0.50	0.00	0.00	0.00	200
E1-OMV-V1	0.50	0.00	0.00	0.00	**155**
**E1-STM-V2**	**0.88**	**0.84**	**0.94**	**0.89**	220
E1-OMV-V2	-	-	-	-	-

**Table 18 sensors-21-01031-t018:** Results of the networks that were integrated into the embedded system. The best experiment is marked in bold.

ID	Accuracy	Precision	Recall	F1	Matrix Size (Kb)	RAM (Kb)
E2-(STM&PC)-V1-025-96	0.50	0.00	0.00	0.00	237,320	55,296
E2-(STM&PC)-V1-025-128	0.40	0.25	0.10	0.14	237,320	98,304
**E2-(STM&PC)-V2-010-96**	**0.88**	**0.84**	**0.94**	**0.89**	**213,184**	**138,240**

**Table 19 sensors-21-01031-t019:** Results of the networks that were not properly integrated into the embedded system. These results were only tested on a PC, as memory restrictions impeded the integration. The best experiment is marked in bold.

ID	Accuracy	Precision	Recall	F1	Matrix size (Kb)	RAM (Kb)
E2-(STM&PC)-V1-025-224	0.43	0.40	0.28	0.33	237,320	301,056
E2-(STM&PC)-V2-015-96	0.91	0.92	0.90	0.91	281,144	138,240
**E2-(STM&PC)-V2-035-96**	**0.95**	**0.94**	**0.96**	**0.95**	**611,912**	*
E2-(STM&PC)-V3-96	0.92	0.92	0.92	0.92	1,227,584	*

**Table 20 sensors-21-01031-t020:** Results of MobileNet V2 integrated into STM32-H747i-Disco using different quantisation techniques. The best experiment is marked in bold.

ID	Accuracy	Precision	Recall	F1	Matrix Size (Kb)	RAM (Kb)
E3-V2-QAT	0.50	0.00	0.00	0.00	130,632	138,240
**E3-V2-ATQ**	**0.88**	**0.84**	**0.94**	**0.89**	**213,184**	**138,240**

## Data Availability

The data used for training the networks have been extracted from Visual Wake Words dataset. The subset used for testing is available in https://github.com/Gorospe/tf_person_classifier.

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
