# Peer review of "A Generalization Performance Study Using Deep Learning Networks in Embedded Systems"

_sensors, 2021, doi:10.3390/s21041031_

Round 1
Reviewer 1 Report
This paper presented a generalization approach to use a DL algorithm in embedded system. The selection of neural network parameters and the influence on network accuracy were discussed. Three different platforms were used to carry out the experiments. The article was clearly described and the experiments were described in detail. It is a topic of interest to the researchers in the related areas but the paper needs some improvements before acceptance for publication. Here are some suggestions for the major revision.
- Please check that all abbreviations were written in full spelling when they first appear, such as PC.
- Figures need to be clearer, and it is recommended to label the subtitle in the figures, e.g., Figure 2 and Figure 4.
- This paper mainly focuses on the experimental comparison of Mobilenet Is it necessary to compare with other series of networks? For example, SqueezeNet. By the way, SqueezeNet in Table 2, is there one more ‘s’?
- The format of the tables should be consistent, and three-line format is recommended.
- Please use the same font size in different figures, such as Figure 1 and Figure 8.
- Deep learning is widely applied in various fields of engineering. The authors are encouraged to cite more up-to-date articles to improve the integrity of the introduction (Recognition and localization methods for vision-based fruit picking robots: a review; Vision-based three-dimensional reconstruction and monitoring of large-scale steel tubular structures).
Reviewer 2 Report
I like the study.
Please improve the paper's conclusions.
I would like to see more for future work.
Reviewer 3 Report
This paper proposes a development environment based on MBed OS and Tensorflow Lite to use deep learning architectures/models in micro-controllers and any general-purpose embedded system (i.e. helps DL training and inference on edge devices). The paper focuses on an important research problem considering the current trend to move AI computing from the Cloud to the Fog and the Edge and it comes with an appropriate performance analysis of the proposed platform. Overall, the proposed contribution sounds technically and scientifically. However, there are some drawbacks and weaknesses which need to be resolved in a revised version of the manuscript:
1- The quality of the presentation of the paper needs to be improved. The paper doesn't flow well and at some points, it is hard to read and follow. The sections are not well connected.
2-Challenges need to be highlighted in the Introduction, the significance of the work needs to be better discussed.
3-While authors provided a piece of comprehensive background information on the state of the art for DL, EC, Fog, etc, there is still a missing section on Related Work discussing the most relevant works and similar papers aimed to bring DL training/inference into embedded systems.
4-The structure of the paper needs to be significantly improved, separation of the results section from the discussion sections can lead to confusion. The current structure is more like a dissertation/thesis structure. The subsection "3.1. Introduction" introduces some inconsistency with regards to other sections. The results are presented to the readers in section 5.Results without proper descriptions/discussion of results and readers should move to the next section 6. Discussion to see this information. Description/discussion of the results should be presented next to the result tables.
5-Figures 8, 9, and 10 might be improved with better diagrams and visualization of results.
6-While the major contribution of this paper seems to be a development environment for DL integration into micro-controllers, the authors lack to provide sufficient technical details on their proposed platform, rather they focused on evaluation implementation of different DL networks. Although these evaluations cant help to understand the position of the proposed integration versos similar proposals and other related works in the current literature. In addition to further technical details on integration and development environment, the paper needs to adequately discuss challenges and justify the significance of the proposed contribution. I suggest using a big picture to demonstrate components of the environments, their relationships, and challenges.
7-The paper lacks to properly discuss/prove how the proposed environment can be generalized for "a wide variety of embedded systems"
Round 2
Reviewer 1 Report
ACCEPT
Reviewer 3 Report
Authors have properly applied all my comments in the revised version of the manuscript and the manuscript has been significantly improved.